# Grating Assembly Dissected in Periodic Bands of Poly (Butylene Adipate) Modulated with Poly (Ethylene Oxide)

**DOI:** 10.3390/polym14214781

**Published:** 2022-11-07

**Authors:** Chia-I. Chang, Eamor M. Woo, Selvaraj Nagarajan

**Affiliations:** Department of Chemical Engineering, National Cheng Kung University, No. 1, University Road, Tainan 701-01, Taiwan

**Keywords:** crystallization, periodic grating assembly, iridescence crystals, interior dissection, microbeam X-ray

## Abstract

Polarized optical microscopy (POM), scanning electron microscopy (SEM), and synchrotron microbeam wide-angle X-ray diffraction (WAXD) were used to investigate the mechanisms of periodic assemblies leading to ring-banded crystal aggregates with light-grating capacity for iridescence in poly (1,4-butylene adipate) (PBA) modulated with poly (ethylene oxide) (PEO). A critical finding is that the PBA crystal assembly on the top surface and in the interior constitutes a grating architecture, with a cross-bar pitch equaling the inter-band spacing. The inner lamellae are arranged perpendicularly to the substrate under the ridge region, where they scroll, bend, and twist 90° to branch out newly spawned lamellae to form the parallel lamellae under the valley region. The cross-hatch grating with a fixed inter-spacing in the PBA aggregated crystals is proved in this work to perfectly act as light-interference entities capable of performing iridescence functions, which can be compared to those widely seen in many of nature’s organic bio-species or inorganic minerals such as opals. This is a novel breakthrough finding for PBA or similar polymers, such as photonic crystals, especially when the crystalline morphology could be custom-made and modulated with a second constituent.

## 1. Introduction

There are various morphologies of spherulites in polymer crystallization. For example, there are dendritic spherulites, ring-band spherulites, Maltese-cross ringless spherulites, and so on. According to the optical properties of bands, the ring-banded spherulites can be divided into non-birefringent ring-banded spherulites, single ring-banded spherulites [1], double ring-banded spherulites [2], and so on. Although the ring-banded spherulites have been investigated in different morphologies under POM, the mechanism of ring-banded spherulites attracts many researchers trying to explain it. Around the 1950s, many scholars [3] proposed that the ring-banded spherulites would form periodic lamellar arrangements because of the continuous twisting mimicking a DNA spiral. These scholars have proposed screw dislocation [4,5], the effects of concentration and thermal or mechanical fields [6,7], unbalanced surface stresses [8], etc., to illustrate that the lamellae will produce spirals. However, it is still not possible to fully prove and explain why the periodic spiral width observed under TEM and the ring-banded width observed under POM of polyethylene (PE) spherulites are different [3,9,10]. To explore the formation mechanism of ring-shaped spherulites, in 2012, Woo et al. [11] prepared poly (ethylene adipate) (PEA) in bulk specimens crystallized at T_c_ = 28 °C and analyzed its interior lamellar assembly by SEM/AFM microscopy. The lamellae are arranged in onion-like layers, and there is no observation of continuous spiral or helical twisting. In 2014, Lugito et al. [12] found that the lamellae are arranged perpendicular to each other in the interior layer of PEA ring-banded spherulites. So a new mechanism is proposed: the corrugated-board model. This novel point of view is even more convincing in explaining that the ring-banded interior lamellae of PEA alternately grow periodically to form ring-shaped spherulites, which grow in the tangential direction and then turn to the radial direction. In addition to PEA, many polymer systems can find the morphologies of ring-banded spherulites, and many studies have repeatedly pointed out that there are many doubts about the mechanism of continuous lamellar twisting in the past. For example, in 2017, Tu et al. [13] observed that among the ring-banded spherulites of poly (nonamethylene terephthalate) (PNT), the interior fan-like lamellae branch to the right and form upwards like a fan. In a poly (L-lactic acid) (PLLA)/poly (ethylene oxide) (PEO) (50/50) system [14] crystallized at T_c_ = 110 °C, ring-banded PLLA spherulites are proved that the interior lamellae are composed of edge-on and flat-on alternatively and mutually to form discontinuous interfaces. Other than localized lamellar bending and twisting to fit into a compact space, most periodic assemblies seen in polymer spherulites are not rationalized by models of continuous helix-twist of single-crystal plates initiating from a common nucleus center.

Poly (1,4-butylene adipate) (PBA) is an aliphatic semicrystalline polyester that has received intensive investigations. Due to its lack of tough polymer chains such as benzene in its chemical structure, the mechanical strength/modulus and the melting point of PBA is low. In applications, owing to the low stability of neat PBA itself, it is often incorporated into other plastics by blending or copolymerizing with other polymers to form final end-use products. For example, Showa-Denko Corporation in Japan has a product called Bionolle [15], a material developed by a copolymer of PBA and poly (butylene succinate) (PBSu), which is widely used in packaging and plastic bags. The characteristics of degradation also reduce the impact on the environment, further achieving the friendly environmental concept. In addition to its application, PBA is a focus of research in the polymer academia because of polymorphism and its ring-banded spherulites in a specific crystallization temperature. One of the crystal lattices is a monoclinic α-form crystal lattice; the other is an orthorhombic β-form crystal lattice. When the crystallization temperature of PBA is lower than 27 °C, the crystalline phase forms completely into a β-form lattice. However, the crystallization temperature is higher than 32 °C, and the crystalline phase is packed only in an α-form lattice [16]. In the temperature range of T_c_ = 28–31 °C, the crystal lattice of PBA spherulites coexist in α- and β-form. Interestingly, in the temperature range of 28–31 °C, the ring-banded crystalline morphology of PBA is exactly formed, while there are no ring-banded spherulites out of this range. The occurrence of polymorphism coincides with ring-banded morphology, which has led scholars to study the relationship between polymorphism and ring-banded morphology. Many researchers conducted work on the thermal behavior of α- and β-form by mixing or copolymerizing PBA with different polymers, such as PBA/poly (vinyl phenol) [17], PBA/poly (vinylidene fluoride) [18], etc.

In 2011, Liu et al. [19] used PBA with different molecular weights to investigate the difference in the characteristics of forming ring-banded spherulites. They found that the formation of ring-banded spherulites by PBA is more affected by their molecular weight and crystallization temperatures. In 2012, Wang et al. [20] used PBA to blend with poly (benzyl methacrylate) (PBzMA) and poly (phenyl methacrylate) (PPhMA), respectively, to prove that there is no direct relationship between the formation of PBA ring-banded morphology and crystal-lattice polymorphism. In addition, in 2005, Wu et al. [21] investigated the α-form nucleus formed by PBA at a crystallization temperature of 32 °C by heat treatment at different elevated temperatures. It is found that the α-form PBA has a crystal-memory effect, which can form α-form crystal lattice below T_c_ = 27 °C without being affected by the crystallization temperature. In conclusion, most scholars pay attention to the crystal lattice of PBA, and the interior lamellar assembly is less discussed. However, in 2008, Frömsdorf et al. [22], using atomic force microscopy (AFM), found that continuous helix-twist lamellae of the ring-band spherulites were not observed; yet, that work was limited to 2-D observations. In 2015, Lugito et al. [23] investigated the 3D crystalline morphology blend of two polymers, PBA and PEA, which can form ring bands in the same temperature windows. They found that the different crystallization rates of PBA and PEA not only cause the PBA/PEA blend system to form a nest-like morphology in a specific composition but also can make cavities in the interior structure of ring-banded spherulites in different compositions at the crystallization temperature of 30 °C. In addition, Nagarajan et al. [24] investigated the cross-section of fracture-interior surfaces of PBA. They reported that although the morphology contrast was not clear in neat PBA (without diluents), the interior lamellar arrangement of the ring-banded PBA spherulites at T_c_ = 30 °C is perpendicular to each layer from the circumferential and radial direction. The interior lamellar arrangement in the circumferential direction corresponds to the ridge on the top surface, and the interior lamellar arrangement in the radial direction corresponds to the valley on the top surface. Between the circumferential and radial layers in the interior structure, the circumferential lamellae are twisted and bent at a nearly 90 ° angle to branch and merge into the radial lamellae. Moreover, the thickness of circumferential and radial layers exactly matches the optical band in POM.

In earlier work on the morphology of neat PBA [24], it was proved that neat PBA was capable of packing into ring bands; yet, the orderliness or regularity of aggregates from neat PBA is a bit away from perfect. Relevant investigations on other polymers have shown that the crystallization of polymers in the presence of suitable diluents as impurities may induce more distinct and regular order in the periodicity of growth [25,26,27]. This work aimed to extend the depth and scope of the analysis in the previous work [24]. An earlier work on analyzing ring-banded poly (ethylene adipate) (PEA) aggregates crystallized at T_c_ = 28–30 °C proved that they possess grating architectures composed of cross-hatch lamellae with mutually perpendicular intersections, where the dimensional scales of the interior crystal assembly displays bio-mimetic structures suitable for iridescent functions [28]. Some semi-natural cellulose crystals or self-assembled non-crystal wrinkles in manmade block copolymers or liquid crystals [29,30,31,32,33,34,35,36] were found to possess iridescence-reflection capacities similar to those commonly seen in some natural bio-crystals. Photonic crystals in the form of periodic grating structures are commonly seen in nature [37,38,39,40,41].

Thus, our preliminary work amply disclosed that the clarity of morphological contrast highly depended on diluents chosen to co-crystallize with PBA. Furthermore, suitable diluents in PBA can regulate and modulate the growth architectures of periodicity and enhance the contrast of interior crystal aggregation, especially in cases where the diluents in crystallized PBA aggregates can be conveniently etched out without altering the original assembly of PBA crystals. Water-soluble polymers such as poly (ethylene oxide) (PEO) would be ideal candidates because the PBA crystals are not water-soluble. A combination of the collective evidence of the interior structure and optical banded spacing can sufficiently prove the lamellar assembly mechanism of PBA ring-banded spherulites; however, it does not provide a clear enough interior fracture surface to explain the mechanism of how the interior lamellae are twisted and bent. In this study, better-contrast interior fracture-surface morphologies in PBA crystallized with PEO as a diluent were investigated to explain how the interior lamellae twist and branch to form different orientations. Moreover, the correlation between the lamellar assembly of the top surface and the interior structure is connected by observing fracture surfaces. Periodic crystal aggregates with 3D architectures could be custom-tailored for probing iridescence properties for light-interference applications. A novel 3D growth mechanism of ring-banded spherulite is proposed.

## 2. Experimental

### 2.1. Materials and Preparation

PBA and PEO were dissolved into chloroform and made into a 2 wt.% polymer solution. PBA was purchased from Aldrich Chemical Company, Inc. (St. Louis, MO, USA), with M_w_ = 12,000 g/mol, T_g_ = −66.2 °C, and T_m_ = 56~60 °C. PEO was purchased from Aldrich Chemical Company, Inc. (St. Louis, MO, USA), with M_w_ = 200,000 g/mol, T_g_ = −60 °C, and T_m_ = 64 °C. The solution was cast on a glass slide at 35 °C then executed in a vacuum oven for one day to remove the solvent. Samples were heated to maximum melting temperature (T_max_ = 180 °C) for 2 min on top of the hotplate to erase the thermal history, then quickly removed from the hotplate to a microscopic hot stage being preset at a specific crystallization temperature (T_c_) and held there till full crystallization. In addition, all the fully crystallized samples were later etched with methanol to remove the PEO constituent from the PBA crystalline architectures to facilitate the analyses of SEM and WAXD.

### 2.2. Apparatus

Polarized-light optical microscopy (POM). Crystallization and characterizing the crystalline morphology of the PBA/PEO blend system were observed with a polarized-light microscope (Nikon Optiphot-2, POM, Tokyo, Japan) equipped with a Nikon Digital Sight (DS-U1) digital camera, CCD digital camera (NFX-35) and a microscopic heating stage (Linkam THMS-600 with a T95 temperature programmer). A λ-tint plate (530 nm) was inserted in POM to make contrast interference colors for all POM graphs. In addition to the regular lens, a 100× objective lens was used to observe specimens at magnifying 1000 times. To utilize the 100× objective, the oil was dropped on the back of the sample, and the 100X objective lens was moved to make intimate contact with the oil.

High-resolution field-emission scanning electron microscopy (HR-FESEM). Samples were examined and characterized using high-resolution field-emission scanning electron microscopy (Hitachi SU8010, HR-FESEM, Tokyo, Japan) for the detailed top surface and interior lamellar arrangement. Samples, after proper fracturing with etching and drying, were coated with platinum using vacuum sputtering (10 mA, 300 s) prior to SEM characterization.

Synchrotron microbeam wide-angle X-ray diffraction (WAXD). The facility was located at National Synchrotron Center (NSRRC, Hsinchu, Taiwan). Source: IU22, detector: Dectris Eiger 1M, detector distance: 0.082 m, microbeam size: 16 μm, photon energy: 15 keV, wavelength: 0.8263 Å, q-range: 0.1~3 Å^−1^. The specimens were prepared in a solution-casting method, in which a droplet of the solution was cast on the PI film. The solvent was first evaporated in air, then completely removed by placing it in a vacuum oven for one day. Afterward, the specimens were subjected to proper heat treatments to crystallize completely at specific T_c_’s, and used methanol to remove the PEO constituent from the crystallized film specimens. Then the well-prepared specimens were transported to synchrotron X-ray microbeam to obtain the WAXD signals of the specimens, whose 1D/2D crystal lattices at specific micro-domain spots were analyzed.

## 3. Results and Discussion

### Crystalline Morphologies of PBA/PEO Blends

The inclusion of PEO into PBA was aimed at using PEO as a modulating agent for controlling the regularity of periodic bands in PBS spherulites. Furthermore, PBA is not soluble in water or methanol; yet, PEO is methanol- or water-soluble·, which can be easily etched out of PBA architectures without disrupting the crystal frames. From a preliminary study, it was known that neat PBA develops the most regular ring bands when crystallized at T_c_ = 28–31 °C. With the inclusion of the PEO constituent at a suitable composition into PBA to form PBA/PEO blends to be crystallized at controlled T_c_ = 28–34 °C, the best parameters were chosen for the suitable temperature window to develop the ideal ring bands for convenience of analysis. Prior to isothermal crystallization, PBA films were soaked at a maximum molten temperature (designated as T_max_) to erase prior thermal history. Note that the maximum molten temperature would influence the regularity of ring bands even if PBA is crystallized at the same Tc.

From the preliminary results, T_max_ = 180 °C (held for 2 min) was chosen and fixed for preparing samples for analyzing the PBA banded crystals. Figure 1 shows the PBA morphology with respect to variables of Tc and compositions. The row of graphs shows the kinetic variables being compositions (PEO contents in PBA/PEO blends), and the column of graphs shows Tc varying from 28 to 34 °C. Obviously, the combination of T_c_ = 30 °C and the composition of PBA/PEO (90/10) yielded the best regularity of bands in PBA. With PEO content greater than 10 wt.% or at Tc outside this specific temperature (30 °C), the lamellae in PBA crystallization do not self-assemble to regular periodicity; instead, highly dendritic patterns or corrupted rings are seen.

Apparently, there is a transition zone from regular ring-banded patterns to highly dendritic ones, and this transition zone is exactly at the "corrupted ring band pattern". This fact suggests that the transformation of crystal aggregation from circular ring bands to straight dendrites usually goes through a zone where the regularity of ring bands starts to disintegrate, where the original phenomenon of lamellae growing in a synchronized pace of radial direction then bend/twist to a circumferential one in periodic intervals, is no longer preserved. Instead, across the transition zone, all lamellae grow much longer at a non-synchronized pace, and periodicity disappears. 

Samples of PBA ring-banded aggregates were prepared by fixing T_c_ = 30 °C with a composition of PBA/PEO (90/10) to obtain the best regularity in ring bands. Note that introducing 10 wt.% PEO in the crystallization of PBA from the PBA/PEO blend did not change the lamellar architecture compared to that in the crystallized neat PBA. Yet, the PEO component could be easily etched off from the crystallized PBA/PEO (90/10) blend, giving much better morphology contrast for convenience of analysis for mechanisms leading to the striking periodic assembly. Figure 2a shows SEM micrographs for the methanol-etched top surface of PBA/PEO (90/10) crystallized at 30 °C. Obviously, the periodic ring bands, as seen in the SEM micrographs, are in agreement with those seen in POM results, except that now one can see clearly that numerous radial-oriented crevices are present in the ridge bands (Figure 2b). By zoom-in to a specific zone of the periodic band, Figure 2b shows that the lamellae on the ridge bands initially grow radially for a distance of ca. 4 μm, then abruptly bend and twist (in a clockwise sense) to merge into a zone marked as valley) (red-color dash curve). Note that the radial-oriented crevices mark the boundaries between the successive radial-oriented lamellar plates on the ridged band. Oppositely, no crevices are present in the valley region, suggesting that the lamellae in the valley regions are dramatically altered in the top surface and interiors, which will be analyzed in the latter sections of interior 3D morphology. The presence of crevices serves as a useful tip for probing the assembly differences in the ridge vs. valley bands on top surfaces and interiors.

Regular POM characterization does not have high enough magnification; thus, oil-contact lenses were utilized for viewing and confirming that the crevice-line band is indeed the protruded "ridge". Figure 3a shows an SEM micrograph for a quarter (1/4) section of a banded PBA spherulite. Figure 3b shows a POM image (with no tint plate) for the same quarter section of PBA spherulite. Obviously, the "brighter band" has a band width of ~4–5 μm, while the darker or "extinction" band is narrower with a band width of ca. 1–2 μm. From the SEM image, the radial-aligned crevices have 4–5 μm in length, and the crevice-free zone has a 1–2 μm gap. Thus, from this direct comparison of SEM and high-mag. oil-contact POM images, it can now be confirmed that in the oil-contact POM image, the brighter band zone is correlated to the radial-aligned crystal lamellae (labeled as the "ridge zone") in the SEM image, while the bent/twisted lamellar tails in the SEM image is correlated to the darker or extinct optical band that is labeled as "valley zone". The brighter or higher optical retardation is due to thicker stacks and more highly oriented radial-oriented lamellae. The darker or nearly extinct optical retardation in the valley zone is due to a lack of or much less orientation and thinner stacks of these crystal tails that bend and twist in the valley zone. 

After analysis of the top-surface relief patterns, it is essential to probe the interiors and how assembly on the top surface is correlated to the architecture of lamellae in 3D interiors. Crystallized specimens of PBA/PEO (90/10) at T_c_ = 30 °C were control-fractured for the best evaluation of the cross sections. Section-view along the tangential fracture is shown in Figure 4a, which displays the periodic assembly of the interior and a top surface for direct correlation. Obviously, crevices are not only on the top surface, as discussed earlier; now, after interior dissection, the crevices are also present at specific spots in the interior cross-section of the ring-banded PBA. However, in contrast to those on the top surface, the crevices in the fractured interior are not only vertical-oriented at one location but also horizontal-oriented at other regions. Regardless of the two different orientations of the crevices, they all exist between the lamellar bundles as distinct boundaries, which suggests that the lamellae in the interiors can be assembled as vertically-oriented (evidenced by vertical crevices) or horizontally oriented (evidenced by horizontal crevices) crystal species. So, a critical question is why do the interior lamellae self-assemble as a grating structure with periodic perpendicular-horizontal intersection? If the top-vs.-interior assembly is inspected, it is strikingly clear that the interior vertical-oriented lamellae are directly underneath the top-surface ridge zone, while the interior horizontal-oriented lamellae are directly underneath the top-surface valley zone. Therefore, it is obvious that as the top-surface bands traverse and alternate from ridge to valley zones, the interior lamellar plates are assembled from a vertical-oriented crystal but rapidly re-oriented, or by spawning new branches, to a horizontally-oriented one. Cycles repeat in the 3D interior till complete drainage of the molten species, with the same mode of cycling on the top surface. Yet, the top surface fibril-cilia pattern should be viewed as the thin peripheral crystal edges of the large interior lamellar plates, as the majority of the crystal plates are submerged beneath the top surface. From the vertical lamellae to the horizontal ones, there is a narrow transition zone where the vertical lamellae, in traversing along the circumferential direction, tilt at downward oblique angles rather than lie horizontally, subsequently bend upward, and finally tilt again at upward oblique angles to merge into vertical lamellae in the neighboring cycle. Repetition of the same cycling manner is seen across the circumferential rim. For a clearer illustration of the interior assembly, Figure 4b shows a simplified scheme for depicting the crystal assembly on the top surface, which is correlated to the 3D interior architecture with periodic cycles of orientation changes of the crystal plates’ self-assembly underneath the top-surface bands. The discontinuity between the lamellar bundled plates assembled on the top surface and those in the interior is obvious, which apparently cannot be interpreted by the conventional models of continuous helix-twist plates radiating and extending from a common nucleus center to the periphery.

For 3D views, it was essential that in addition to dissection along the circumferential direction, analysis should also be performed along a radial fracture line. Figure 5a shows an SEM micrograph of a radial-fractured surface of PBA/PEO (90/10) crystallized at 30 °C. Along the radial direction, a clearly alternate up-and-down height variation is seen on the top surface, where the inter-ridge spacing = 4.5 μm, equal to the inter-band spacing as viewed in POM micrographs discussed earlier. In the interiors, along the radial direction, vertical-oriented lamellae (or their bundles) are located underneath the top-surface "ridge zone"; oppositely, horizontally-oriented lamellae (or bundles) are positioned underneath the top-surface "valley zone". From the vertically-oriented lamellae to horizontally-oriented lamellae in the interior, there exists discontinuity of interfaces; then, attention is directed to the assembly correlating the top surface to the interiors. The interior vertically-oriented lamellar plates are parallel-assembled from the bottom substrate to near the top surface; as they merge upward to the top surface to form a "ridge zone", their top edges appear as parallel-aligned fibril cilia, tilting at an oblique angle, but all pointing in the radial direction. This is why when one views the top surface, only cilia crystals are aligned on the ridge, which is because what can be viewed is the lateral edge of interior lamellae that are vertically oriented. Repetition of the same cyclic assembly is seen across the circumferential rim. Similar to the previous dissection across the circumferential direction, Figure 5b shows a simplified scheme by cutting along the radial direction to depict the crystal assembly on the top surface and interior. The scheme shows a 3D architecture with periodic cycles of orientation changes in the interior crystal plates’ self-assembly underneath the top-surface bands. Similar to the previous dissection across the circumferential direction, the radial-dissected assembly results also demonstrate that the vertically-oriented lamellae are positioned underneath the top-surface ridge zone, while the horizontally-oriented lamellae are positioned underneath the top surface valley zone. Once again, discontinuity is obvious between the lamellar bundles assembled on the top surface as well as those in the interior, and the discontinuity does not support the conventional models of continuous helix-twist plates simultaneously radiating and extending from a common nucleus center to the periphery.

The assembly of crystals in bulkier films of different thicknesses might be altered from that seen in thin films. The preliminary result has indicated that ring bands and crystal assembly tend to be invariable in the same modes as long as the film thickness is 10 μm or thicker. This is well demonstrated in Figure 6a,b, which shows SEM graphs for top vs. fractured-interior of ring-banded PBA/PEO (90/10) crystallized at T_c_ = 30 °C with film thickness = 12 μm, or 40 μm (both specimens were solvent-etched), respectively. The assembly patterns in PBA films of these two different thickness levels appear to be similar in their periodicity, inter-band spacing, and highly asymmetric crystals near or around the nucleus region. Figure 6c is a zoom-in SEM of graph (b) to the nucleus region. The appearance of tri-cracks near the nucleus region is typical in most spherulites. For the ring-banded assembly, the lamellae near the nucleus region display several distinct characteristics: (a) the nucleus sheaf-like crystals display discontinuity right outside the nucleation core, and (b) the nucleus sheaf-like lamellae grow initially straight for 2–3 μm, then bend sharply at 60–90° angles to a tangential zone, from which radial lamellae are spawned and then bend to a tangential direction to initiate cyclic packing of ring bands. There is a distinct boundary (narrow-gap crevice) between the radial lamellae and the tangential one, as marked by a pink-color dashed line.

Zoom-in to the radial-fractured interior was attempted for greater details of crystal assembly leading to the notable periodic architectures that display optical bands. Figure 7 shows an SEM graph of the radial-fractured interior of PBA/PEO (90/10) crystallized at 30 °C. The periodic assembly in the interior differs subtly from that on the top surface. On the top surface, there is alternate ridge-to-valley height variation, but in the interior, there is no alternate height variation; instead, periodicity is marked by radial growth followed by tangential bending/twisting into a collective narrow zone. The next crystal growth cycle repeats by the same mode of radial growth followed by tangential bending/twisting into a collective narrow zone. Red arrows in the SEM graph indicate the zone where the radial lamellae sharply bend, twist and merge into a common tangential line. Interior grating cross-bar pitch was estimated from the periodic grating pattern of the SEM graph to be 5.5 μm, which is exactly equal to the optical inter-band spacing as viewed in POM micrographs shown earlier. Red arrows mark the sharp tail-bending/twisting of the interior lamellae upon reaching the radial-tangential interface. The radial lamellae are vertically oriented concerning the substrate plane; once they sharply twist/bend into the narrow tangential zone, they become horizontally oriented with respect to the substrate plane. Furthermore, all tangential lamellae are stagger-stacked in a fish-scale pattern. The radial lamellae are parallel to each other and mostly straight before sharply bending/twisting to merge into the tangential zone.

Synchrotron-source microbeam WAXD was performed on the ring-banded PBA for discerning possible local changes of crystal lattice polymorphism. Figure 8 shows 2D WAXD patterns at various spots in a PBA-banded spherulite crystallized from PBA/PEO (90/10) at 30 °C. Inset is a POM graph for visually illustrating where spots of microbeam X-ray analysis were directed. A microbeam of 15 μm (marked as black dots) was directed to various spots across the banded cycles of different spherulites (Inset on Lower Right). The 2D patterns are similar, but differ in the relative intensity of the two crystal lattices. 

For a better appreciation of relative intensity changes, Figure 9 shows 1D WAXD diffractograms for spots-1 to spot-6, where the intensity of the α-form is relatively constant for all spots. However, on Spots 1, 2, 3, and 5, the intensity of the β-form is weak to nil. The signal for the β-form is discernible only on Spots 4 and 6. The microbeam result confirms that at T_c_ = 30 °C, the banded PBA may be composed of both α and β-form polymorphism crystal lattices. However, the statistical averages of microbeam WAXD show that the presence of β-form is obvious in a fraction of the banded spherulites; in another majority of banded PBA ones, they may still be composed of dominantly the α-form. That is to say that polymorphism is not a necessary condition for packing into periodically banded crystal aggregates.

To correlate the optical bands of alternate birefringence rings with the top-surface morphology and dissected interior as revealed in the SEM analysis, Figure 10a shows a POM graph with alternate orange/blue birefringence stripes for PBA banded spherulite crystallized from PBA/PEO (90/10) at 30 °C. Figure 10b is a schematic summarizing the earlier morphology analysis displaying that the narrow lateral edges of the lamellar plates may naturally appear as "fibril cilia", but a much larger portion of the crystal plates are submerged underneath the top surface, which are almost vertically oriented but tilt at a minor oblique angle with respect to the substrate or top surface. Again in the radial dissection shown and discussed earlier, similar to those discussed earlier for the circumferential dissection, in traversing from the vertical lamellae to horizontal ones, there is a narrow transition zone where the vertical lamellae, in traversing along the circumferential direction, tilt at downward oblique angles, then lie horizontally, and subsequently bend upward, finally tilting again at upward oblique angles to merge into vertical lamellae in the neighboring cycle. To continue, Figure 10c is a schematic showing a summary correlation between the optical bands and interior lamellar assembly. The alternate birefringence color switch indicates that the orientation of the lamellae in the interior, as well as on the top surface, undergoes a cyclic radial-to-tangential change. The interior lamellae form a periodic grating assembly with a fixed cross-bar pitch (5.5 μm), which matches with optical inter-band spacing (5.5 μm) discerned in POM graphs. Note that the crystal assembly in the narrow tangential zone is not as well-aligned as in the radial zone. Furthermore, the lamellae in the tangential zone undergo sharp twisting, bending, and fish-scale-like stagger stacking. Therefore, the optical birefringence in this narrow zone (blue-color zone in Figure 10a) may display weaker retardation or sometimes result in optical extinction (magenta color with a tint plate) if the alignment of the lamellae becomes diminished or more corrupted, especially for the PBA specimens crystallized at T_c_ > 30 °C. 

## 4. Iridescence of Periodically Assembled Crystals

The banded PBA crystal aggregates are structured as a cross-hatch grating made of tangential-to-radial lamellae almost perpendicularly to each other, where the cross-bar pitch of the crystal assembly is equal to the optical inter-band spacing (5.5 μm). Naturally, such an orderly grating assembly with hierarchical lamellar cross-bar pitches ranging from nano- to micrometer scales in the ring-banded PBA is similar to many photonic crystals widely seen in bio-species for light interference, and one may expect that such crystal architectures are capable of performing similar functions. These expectations, however, need experimental justification. Many of nature’s crystal assemblies are known for bio-photonic structures that produce light interference/diffractions [41,42,43,44,45,46]. The interior periodic arrays in either PEA or PBA ring-banded spherulites can also be drawn as an analogy to those in beetles’ exoskeletons with a photonic structure earlier reported by Bartl et al. [47]. In addition to the animal kingdom, plants may display similar functions. Pollia condensata (a striking blue-color berry fruit) is known to possess spiral-stacked shells in the inner walls of the skin cells, which are reportedly responsible for its strikingly bright-blue structural coloration. The nature-evolved cellulose micro-fibrils in spiral-stacked grating architectures result in the intense coloration created by multiple Bragg reflections [39].

Non-grating assembly in ringless PBA aggregates was first assessed for photonic behavior. Figure 11 shows iridescence experimental results for corrupted rings or totally ringless PBA crystal aggregates. Parameters were changed by fixing T_c_ at 32 °C, but the PEO content was adjusted from 10 to 50 wt.% in the PBA/PEO blend. This parametric adjustment at this fixed T_c_ = 32 °C led to the gradual disappearance of original ring bands: (a) at 10 wt.% PEO, the PBA ring bands become highly corrupted and irregular; (b) at 25 wt.% PEO, the PBA aggregates become completely ringless packed with fan-like dendrites; (c) at 50 wt.% PEO, the PBA aggregates remain completely ringless but with coarser dendrites. Figure 11d–f proves that the corrupted/irregular ring-banded or completely ringless PBA aggregates do not display iridescence interference.

By contrast, at a lower T_c_ = 30 °C, with the compositions of the PBA/PEO blend being the same at 10, 25, and 50 wt.% of PEO, respectively, all PBA aggregates are orderly ring-banded, as shown in Figure 12a–c. Figure 12d–f shows that the well-orderly ring-banded PBA aggregates display striking iridescence interference. For comparison, the color photos of Figure 12g–i are three representative natural iridescent crystals: opal, Pollia berry, and nacre (abalone) inner-shell, respectively, which are composed of aligned SiO_2_ micro-crystalline spheres, cellulose fibrils, and CaCO_3_ aragonite/keratin in respective grating assemblies, for performing effective light interferences. By comparison, the iridescence color patterns (Figure 12d–f) in the PBA periodically banded crystals are strikingly similar to those in natural photonic crystals (Figure 12g–i).

To further prove that it is the nature of the periodic banded micro-architectures rather than the macro-shapes of the specimens that yielded the interference iridescence, experiments were conducted on PBA specimens cast in various geometrical shapes. Instead of spreading the specimens as a circular shape on glass slides for iridescence experiments, the specimens were stenciled into a letter "N" prior to crystallization at T_c_ = 30 °C. Figure 13 shows the photonic iridescence of PBA banded crystal specimens stenciled into the letter "N" and crystallized at various T_c_’s from 25 to 35 °C but from a fixed PBA/PEO = 90/10 blend composition. As shown on the lower POM graphs, the morphology patterns vary with T_c_ from ringless at T_c_ = 25 °C to orderly ring-banded at T_c_ = 30 °C and then back to ringless (large dendritic) at T_c_ = 35 °C. Correspondingly, the presence or absence of orderly periodic grating structures governs the photonics and interference features; only the clearly ring-banded PBA aggregates crystallized from PBA/PEO = 90/10 (wt./wt.) at T_c_ = 30 °C display distinct iridescence. The iridescence property is sensitively correlated to the appearance or disappearance of the periodicity in the crystal architectures. Similar to the effect of composition (PEO content variation), the crystallization temperature influences the morphology of the PBA aggregates. Both composition and T_c_ can be utilized for adjusting and modulating (i.e., custom-tailoring) the PBA crystal aggregates into desired grating architectures for the best performance of iridescence properties.

## 5. Conclusions

This work proposes a novel mechanism of interior lamellar grating-like assembly in the periodic ring-banded PBA crystals, which have the potential to be custom-tailored to perform iridescence light interference. PBA can be modulated with a diluent for controlling periodicity and regularity in its crystallized crystals to be irregular assembly or periodically ring-banded crystal aggregates with different levels of orderliness in assembled architectures. During crystallization growth, PBA can be modulated with a diluent for controlling periodicity and regularity in its final crystallized crystals to be either irregular assembly or periodically ring-banded crystal aggregates with different levels of orderliness in assembled architectures. To correlate between the top surface and interior-dissected structures, the inner lamellae are arranged perpendicularly to the substrate under the ridge region, where they scroll, bend, and twist 90° to branch out newly spawned daughter lamellae to form the parallel lamellae under the valley region; subsequently, cycles self-repeat in the same manner till final maturing impingement. Thus, the crystal assembly on the top surface and in the interior constitutes a grating architecture, with a cross-bar pitch equaling the inter-band spacing. The cross-bar pitch and microstructures of assembly could be controlled by T_c_, type or composition of diluents, etc. The cross-hatch grating with a fixed inter-spacing in the PBA aggregated crystals, with proper modulation as demonstrated in this work, acts as light-interference entities, which perform similar iridescence functions as those widely seen in many nature’s organic bio-species or inorganic minerals such as opals.

This is a novel finding with significant practical iridescence applications for PBA or similar polymers to function as photonic crystals, especially when the crystalline morphology could be custom-made and modulated with a second constituent. PBA crystals made with corrupted/irregular ring-banded or completely ringless patterns do not display iridescence interference at all. By contrast, as PBA crystallization assembly is modulated with a suitable T_c_ and diluent’s composition, the final PBA periodically banded crystal aggregates are capable of displaying iridescence color patterns strikingly similar to those of natural photonic crystals such as the iridescent crystals opal, Pollia berry, and nacre, each with distinctly subtle different grating assemblies but sharing commonality in performing light interferences. 

## Figures and Tables

**Figure 1 polymers-14-04781-f001:**
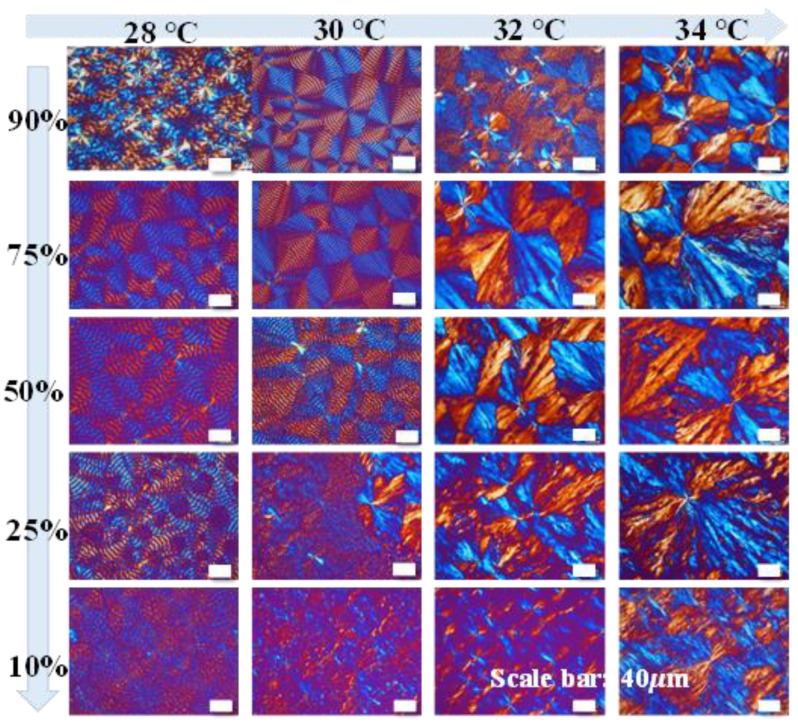
POM graphs of PBA/PEO blend of various compositions (PEO = 10–90 wt.%, top to bottom rows) crystallized at different T_c_ (28–34 °C, Left to Right columns) by quenching from T_max_ = 180 °C.

**Figure 2 polymers-14-04781-f002:**
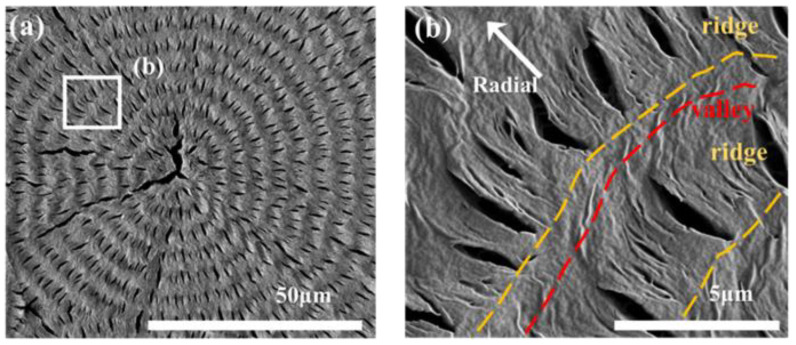
SEM graphs for methanol-etched top surface of PBA/PEO (90/10) crystallized at 30 °C: (**a**) top surface of an entire spherulite, (**b**) zoom-in to lamellae on top surface displaying periodic tail-bending/twisting to the circumferential direction at the valley.

**Figure 3 polymers-14-04781-f003:**
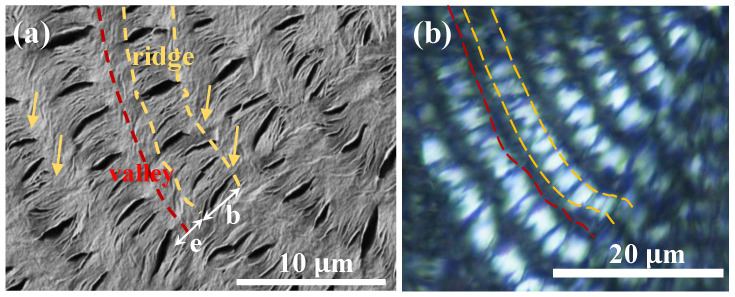
(**a**) SEM graph, and (**b**) oil-contact POM graph (with no tint plate) of top surface of PBA/PEO (90/10) crystallized at 30 °C. [Oil-contact POM at max. magnification of 2000×].

**Figure 4 polymers-14-04781-f004:**
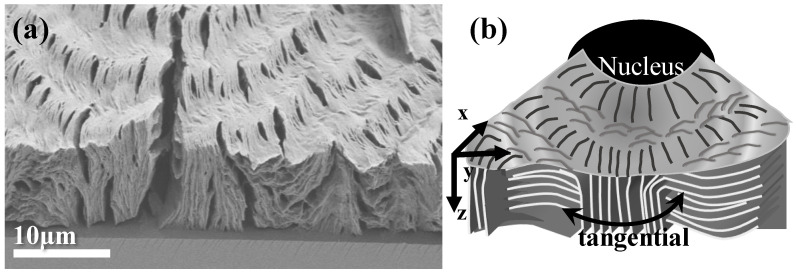
(**a**) SEM graph of tangentially-fractured interior surface of PBA banded crystals crystallized from PBA/PEO (90/10) at 30 °C and (**b**) schematics for correlation between the interior lamellae and top-surface ring bands.

**Figure 5 polymers-14-04781-f005:**
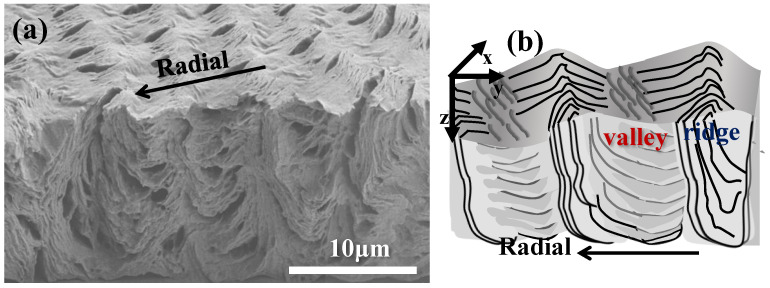
(**a**) SEM graph of a radial fracture surface of PBA/PEO (90/10) crystallized at 30 °C, and (**b**) schematic illustrations for correlation between the interior lamellae and top-surface ring bands.

**Figure 6 polymers-14-04781-f006:**
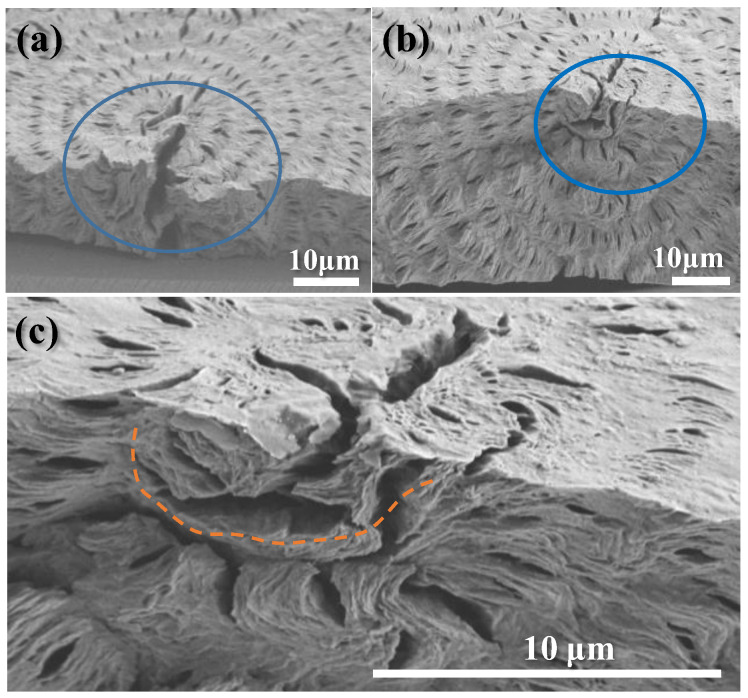
SEM graphs for top vs. interior of ring-banded PBA/PEO (90/10) crystallized at T_c_ = 30 °C with film thickness: (**a**) 12 μm, (**b**) 40 μm, and (**c**) zoom-in of the graph (**b**) to the nucleus region. Blue circles indicating the nucleus zone with cracks.

**Figure 7 polymers-14-04781-f007:**
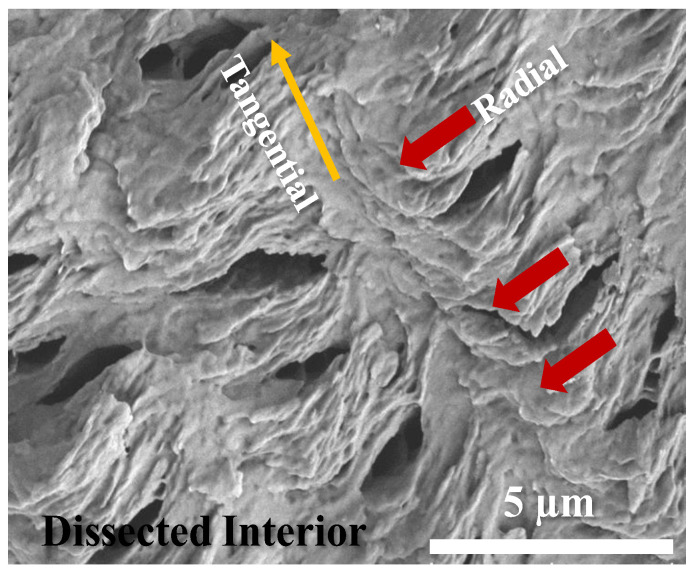
SEM graph of radial-fractured interior of PBA/PEO (90/10) crystallized at 30 °C. Red arrows indicate the zone where radial lamellae sharply bend, twist, and merge into a common tangential line.

**Figure 8 polymers-14-04781-f008:**
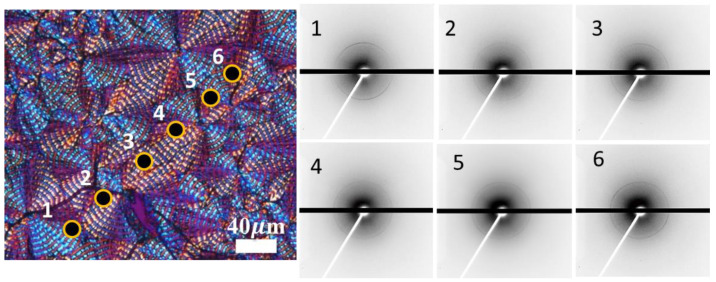
POM graph from PBA/PEO (90/10) at 30 °C marked with various microbeam exposure spots and corresponding 2D-WAXD patterns.

**Figure 9 polymers-14-04781-f009:**
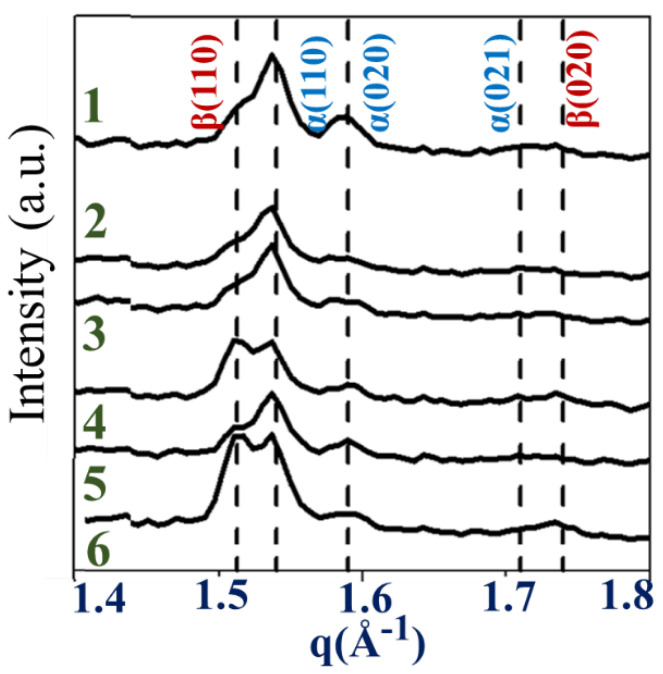
1D WAXD diffractograms at various spots in PBA banded spherulite crystallized from PBA/PEO (90/10) at 30 °C.

**Figure 10 polymers-14-04781-f010:**
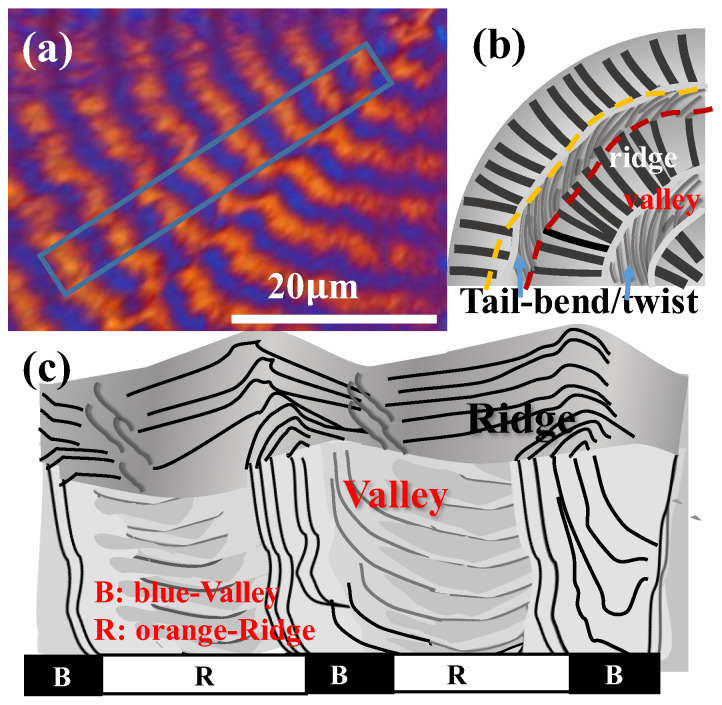
(**a**) POM graph with alternate birefringence bands, (**b**) schematic for top-surface morphology, and (**c**) schematic of correlation between optical bands and interior lamellae of banded PBA crystallized from PBA/PEO (90/10) at 30 °C. R-red tint color, B = blue tint color. Blue-block indicating the radial direction of alternate birefringence changes.

**Figure 11 polymers-14-04781-f011:**
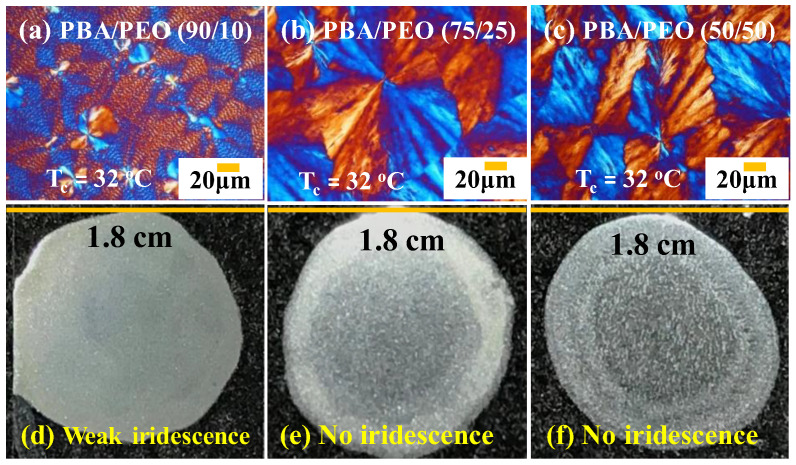
(**a**–**c**) Ringless birefringence patterns as viewed in POM, (**d**–**f**) weak or complete lack of iridescence from corrupted/irregular ring-banded or completely ringless or PBA (T_c_ = 32 °C) crystallized from PBA/PEO blend of three different compositions: 90/10, 75/25, 50/50.

**Figure 12 polymers-14-04781-f012:**
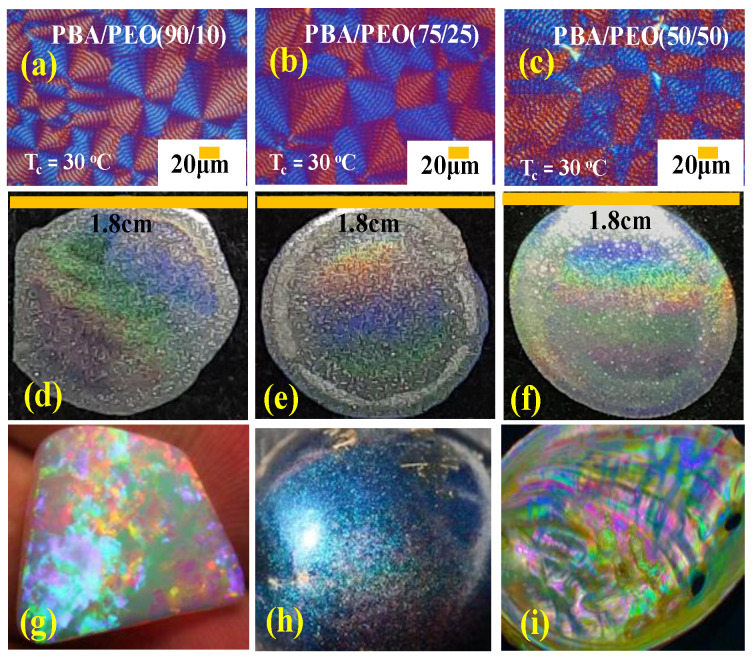
(**a**–**c**) POM micrographs, (**d**–**f**) photonic iridescence from ring-banded PBA (T_c_ = 30 °C) crystallized from PBA/PEO blend of three compositions: 90/10, 75/25, 50/50, (**g**) opal [48], (**h**) Pollia berry [49], (**i**) nacre/abalone shell (self-photo work).

**Figure 13 polymers-14-04781-f013:**
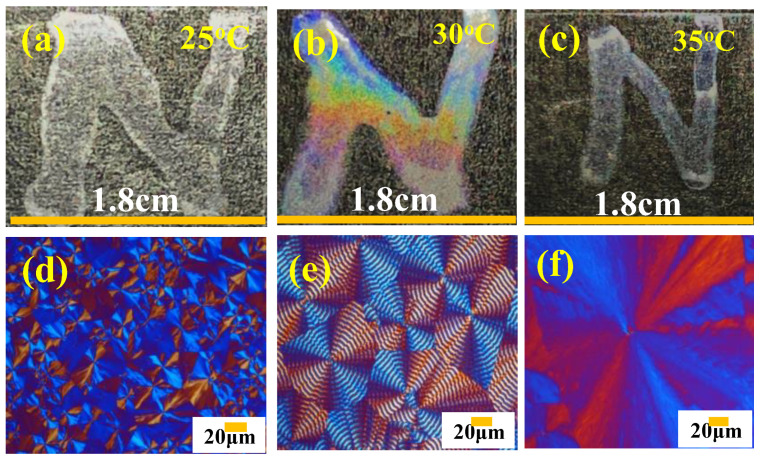
Photonic iridescence of specimens hand-stenciled into a letter "N" and crystallized at various T_c_’s: (**a**) small-size ringless PBA (T_c_ = 25 °C), (**b**) medium-size regularly ring-banded PBA (T_c_ = 30 °C), (**c**) large-size dendritic PBA aggregates (T_c_ = 35 °C), and (**d**–**f**) POM micrographs of corresponding specimens cast into "N" letter from PBA/PEO (90/10) blend.

## Data Availability

Data are contained within the article and are available upon reasonable request.

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
