# Peer review of "Grating Assembly Dissected in Periodic Bands of Poly (Butylene Adipate) Modulated with Poly (Ethylene Oxide)"

_polymers, 2022, doi:10.3390/polym14214781_

Round 1

Reviewer 1 Report

The manuscript investigates the mechanism to assemble periodic ring-banded crystal aggregates in PBA (Poly Butylene Adipate), modulated with PEO (Poly-Ethylene Oxide). PEO is water-soluble so it can be removed after crystallization, since PBA isn't water-soluble. A first analysis, performed by POM (Polarized Optical Microcopy), establishes the right concentration and crystallization temperature to obtain regular periodicity. Afterward a top surface and a vertical growth characterization are performed with SEM and POM and synchrotron microbeam wide-angle X-ray diffraction. The manuscript demonstrates also that the so obtained periodic ring-banded crystals show iridescence like Photonic  Crystals. The paper is clear and well written and also of interest for the scientific community. In my opinion is almost ready for publication.

Some minor improvements are reported below:

1) Line 338 please double check "10 m". I suppose the "mu" of micron is missing.
2) Figure 7, please double check  "5 m" in the figure legend.

Author Response

Reviewer #1

The manuscript investigates the mechanism to assemble periodic ring-banded crystal aggregates in PBA (Poly Butylene Adipate), modulated with PEO (Poly-Ethylene Oxide). PEO is water-soluble so it can be removed after crystallization, since PBA isn't water-soluble. A first analysis, performed by POM (Polarized Optical Microcopy), establishes the right concentration and crystallization temperature to obtain regular periodicity. Afterward a top surface and a vertical growth characterization are performed with SEM and POM and synchrotron microbeam wide-angle X-ray diffraction. The manuscript demonstrates also that the so obtained periodic ring-banded crystals show iridescence like Photonic Crystals. The paper is clear and well written and also of interest for the scientific community. In my opinion is almost ready for publication.

Authors’ response/revision: We are grateful for referee #2’s appreciation and favorable evaluation.

Some minor improvements are reported below:

Q1) Line 338 please double check "10 m". I suppose the "mu" of micron is missing.

Authors’ Answer/revision: As per the review suggestion, the above words have been modified in revised manuscript R1.

Q2) Figure 7, please double check "5 m" in the figure legend.

Authors’ Answer/revision: Maybe the converted PDF file became corrupted.  But we examine the original word file, the legend “5 mm” is correct on Fig. 7.  The scale bar is 5 μm, but the text of “estimate pitch is 5.5 μm”. Both are correct.

Reviewer 2 Report

The question of how to use the polarized optical microscopy (POM), scanning electron microscopy (SEM), and synchrotron microbeam wide-angle X-ray diffraction (WAXD) to investigate the mechanisms of periodic assemblies is an interesting one. so, I agree with the authors that they are conducting a useful exercise.

In fact, in the submitted paper, in order to perfectly act as light-interference entities capable of performing iridescence functions, the cross-hatch grating with a fixed inter-spacing in the PBA aggregated crystals is proved.

The manuscript is reasonably clear, though needs editing to improve some of the grammar. The work is not especially novel, as the grating assembly dissected in periodic bands of butylene adipate modulated with ethylene oxide has been well described in the literature. This is a minor flaw in the manuscript, and the authors need to make a better argument for its significance.

Author Response

Reviewer #2

The question of how to use the polarized optical microscopy (POM), scanning electron microscopy (SEM), and synchrotron microbeam wide-angle X-ray diffraction (WAXD) to investigate the mechanisms of periodic assemblies is an interesting one. so, I agree with the authors that they are conducting a useful exercise.

In fact, in the submitted paper, in order to perfectly act as light-interference entities capable of performing iridescence functions, the cross-hatch grating with a fixed inter-spacing in the PBA aggregated crystals is proved.

The manuscript is reasonably clear, though needs editing to improve some of the grammar. The work is not especially novel, as the grating assembly dissected in periodic bands of butylene adipate modulated with ethylene oxide has been well described in the literature. This is a minor flaw in the manuscript, and the authors need to make a better argument for its significance.

Authors’ response/revision: We are grateful for referee #2’s appreciation and favorable evaluation.

Reviewer 3 Report

The manuscript entitled “Grating Assembly Dissected in Periodic Bands of Poly(butyl- 2 ene adipate) Modulated with Poly(ethylene oxide)” proposed a novel mechanism of interior lamellar grating-like assembly in 501 the periodic ring-banded PBA crystals, which have potential to be custom-tailored to per- 502 forms iridescence light interference. The proposed work is interesting and will be helpful for future research. The obtained results are satisfying and scientifically justified and significant thus the manuscript is recommended for acceptance without any changes.

Author Response

Reviewer #3.

The manuscript entitled “Grating Assembly Dissected in Periodic Bands of Poly(butyl- 2 ene adipate) Modulated with Poly(ethylene oxide)” proposed a novel mechanism of interior lamellar grating-like assembly in 501 the periodic ring-banded PBA crystals, which have potential to be custom-tailored to per- 502 forms iridescence light interference. The proposed work is interesting and will be helpful for future research. The obtained results are satisfying and scientifically justified and significant thus the manuscript is recommended for acceptance without any changes.

Authors’ response/revision: We are grateful for referee #2’s appreciation and favorable evaluation.